# Cardiac rehabilitation for children and young people (CardioActive): protocol for a single-blind randomised feasibility and acceptability study of a centre-based cardiac rehabilitation programme versus usual care in 11–16 years with heart conditions

Lora Capobianco ![ORCID],[1] Mark Hann,[2] Emma McManus,[3] Sarah Peters,[4] Patrick Joseph Doherty ![ORCID],[5] Giovanna Ciotti,[6] Joanne Murray,[7] Adrian Wells[1,8,9]

For numbered affiliations see end of article.

**Correspondence to**
Dr Lora Capobianco;
Lora.Capobianco@gmmh.nhs.uk

## ABSTRACT

**Background** Congenital heart conditions are among the most common non-communicable diseases in children and young people (CYP), affecting 13.9 million CYP globally. While survival rates are increasing, support for young people adjusting to life with a heart condition is lacking. Furthermore, one in three CYP with heart conditions also experiences anxiety, depression or adjustment disorder, for which little support is offered. While adults are offered cardiac rehabilitation (CR) to support their mental and physical health, this is not offered for CYP.

One way to overcome this is to evaluate a CR programme comprising exercise with mental health support (CardioActive; CA) for CYP with heart conditions. The exercise and mental health components are informed by the metacognitive model, which has been shown to be effective in treating anxiety and depression in CYP and associated with improving psychological outcomes in adult CR.

**Method and analysis** The study is a single-blind parallel randomised feasibility trial comparing a CR programme (CA) plus usual care against usual care alone with 100 CYP (50 per arm) aged 11–16 diagnosed with a heart condition. CA will include six group exercise, lifestyle and mental health modules. Usual care consists of routine outpatient management. Participants will be assessed at three time points: baseline, 3-month (post-treatment) and 6-month follow-up. Primary outcomes are feasibility and acceptability (ie, referral rates, recruitment and retention rates, attendance at the intervention, rate of return and level of completion of follow-up data). Coprimary symptom outcomes (Strength and Difficulties Questionnaire and Paediatric Quality of Life) and a range of secondary outcomes will be administered at each time point. A nested qualitative study will investigate CYP, parents and healthcare staff views of CR and its components, and staff's experience of delivering CA. Preliminary health economic data will be collected to inform future cost-effectiveness analyses. Descriptive data on study processes and clinical outcomes will be reported. Data analysis will follow intention to treat. Qualitative data will be analysed using thematic analysis and the theoretical framework of acceptability.

**Ethics and dissemination** Ethical approval was granted on 14 February 2023 by the Greater Manchester East Research Ethics Committee (22/NW/0367). The results will be disseminated through peer-reviewed journals, conference presentations and local dissemination.

**Trial registration number** ISRCTN50031147; NCT05968521.

### STRENGTHS AND LIMITATIONS OF THIS STUDY

⇒ The study will evaluate a novel cardiac rehabilitation intervention (CardioActive) for children and young people with heart conditions, which includes both mental and physical health support based on evidenced-based psychological theory.

⇒ As the study is feasibility, the sample size is limited and the study is not powered to determine efficacy.

⇒ Data collection will be conducted by researchers blind to participant allocation

## INTRODUCTION

Congenital heart disease (CHD) affects 13.8 million children and young people (CYP) under the age of 20 globally.[1] Survivors of CHD face challenges and comorbidities which impact their ability to function,[2] future employment and progression into independent adulthood.[3][4] CYP with heart conditions have a reduced quality of life and poorer psychological functioning,[5–7] as 41% of CYP with CHD experience significant difficulties adjusting to their heart condition, and

30% experience clinically significant anxiety or depression.[7–13] In addition, overprotection from parents, educators and healthcare staff contributes to reduced physical activity and exercise intolerance.[14–17] This is particularly important as sedentary lifestyles are a risk factor for atherosclerosis, cardiovascular disease, obesity and diabetes.[18]

A possible means of improving physical and mental health outcomes would be to facilitate access to a cardiac rehabilitation (CR) programme that improves healthy behaviours such as exercise and psychological self-management skills. CR programmes provide a unique opportunity to improve physical health, provide education around lifestyle and risk factors, and provide psychological support.[19 20] In the UK, following a cardiac event, adults are encouraged to attend CR programmes, which have been recommended by the National Health Service (NHS),[21] the Department of Health, National Institute for Health and Care Excellence[22] guidelines (CG172, CG94 and NG106) and the British Association for Cardiovascular Prevention and Rehabilitation.[23] In adults, CR programmes have been found to reduce patient mortality, morbidity, hospital readmissions, improve quality of life and be a cost-effective intervention.[24 25] While CR programmes are accessible for adults, equivalent programmes are not routinely available for CYP living with heart conditions.

Previous systematic reviews and meta-analyses of paediatric CR have indicated that paediatric CR programmes improve physical, developmental, cognitive and psychosocial outcomes.[24 26–28] Akamagwuna and Badaly[24] conducted a systematic review of 20 paediatric CR programmes. They found that both home-based and outpatient programmes were associated with improvements in cardiorespiratory outcomes, quality of life and psychosocial functioning. Previous youth CR programmes have predominantly focused on improving physical health outcomes, with less attention given to mental health components, despite 41% of patients struggling with adjusting to living with a heart condition. Tesson *et al*[29] conducted a systematic review of psychological interventions for childhood-onset heart disease and found only two studies focused on interventions for CYP.[30 31] However, the results were disappointing, as they did not lower anxiety, depression[31] or improve quality of life.[30] CR with CYP could benefit from routine exercise programmes combined with psychological input aimed at improving adjustment and mental health outcomes.

Drawing on the literature on physical and psychological effects of CR techniques in CYP[29–32] and informed by theory and research on metacognitive therapy (MCT) in adult CR and mental health[33–35], we have devised CardioActive (CA), a six-session CR package for use with CYP. The duration of the CA programme is briefer than previous programmes where patients have on average received 2.5 sessions per week for 16 weeks.[24 26–28 32] The physical health component has been informed by recommendations for physical activity in CYP with CHD[36–39],

and the latest systematic reviews and meta-analyses which suggest that programmes should include a combination of endurance and strength-based exercises.[29–32] The mental health/adjustment component of CA is informed by the metacognitive model[40–42] of anxiety, depression and stress and focuses on helping CYP to reduce their worry and unhelpful self-monitoring processes. According to the metacognitive model psychological distress and failure to adjust is maintained by a style of thinking termed the cognitive attentional syndrome (CAS). The CAS is characterised by repetitive negative thinking (worry, dwelling), monitoring for threat (ie, checking for symptoms), inflexible attention and engaging with unhelpful coping strategies (ie, avoiding exercise, resting more, poor eating and avoidance). Metacognitive approaches may be more appropriate than cognitive therapy approaches as cognitive therapy uses techniques such as reality testing patient's thoughts, which in heart disease patients are often realistic (ie, 'what if my heart disease gets worse?') and may not be amenable to cognitive therapy techniques.[43] This approach has been usefully applied in adult CR to improve psychological outcomes.[34]

While the metacognitive approach is promising, further research is needed to understand the views and responses to the metacognitive approach delivered as part of CR for CYP. As such, the aim of the proposed study is to develop a CR programme (CA), which incorporates mental health support for CYP and to evaluate the acceptability and feasibility of the programme in comparison to usual care.

## METHODS
### Design

The current study is a parallel single-blind randomised feasibility study in CYP aged 11–16 with a heart condition with 12 and 24 weeks follow-up, comparing routine clinical management (treatment as usual (TAU)) against the CA programme. Qualitative investigation will be embedded within the trial. Preliminary economic data will be obtained to inform the optimum way of evaluating cost-effectiveness in any subsequent definitive trial. See figure 1 for an overview of the trial design according to Consolidated Standards of Reporting Trials (CONSORT) guidelines.[44 45] The recommendations for the Interventional Trials 2013 Standard Protocol Items: Recommendations for Interventional Trials Checklist[46 47] is included in Additional file 1. Figure 2 outlines the schedule of recruitment, interventions and assessment.

### Trial population

The trial population is children and adolescents attending cardiology outpatient services.

### Eligibility criteria

In order to be eligible to take part in the study participants will meet the following inclusion and exclusion criteria:

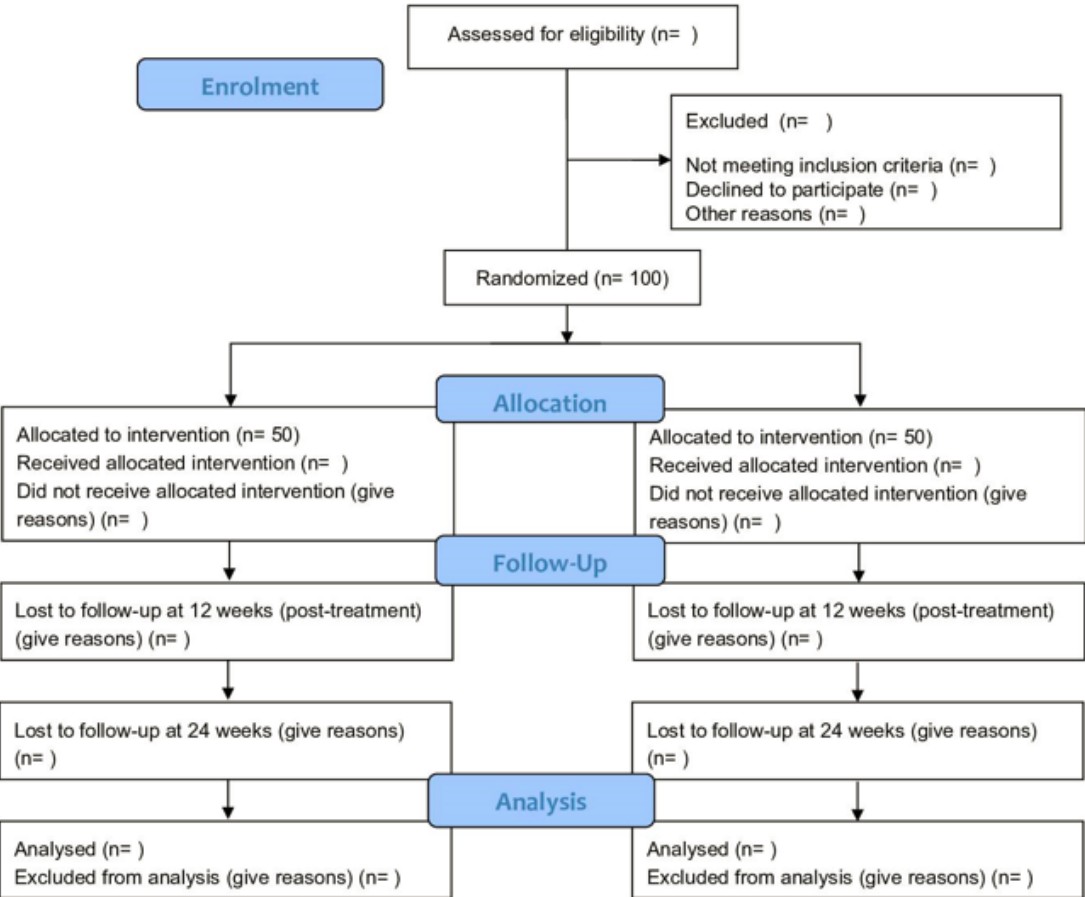

**Figure 1** Trial flow diagram.

## Inclusion criteria

1. Aged between 11 and 16 years.
2. Ability to consent to participate.
3. Native fluency in English language.
4. Diagnosed with at least one of the following: congenital heart disease (all subtypes), cardiomyopathy, cardiac arrhythmia, heart failure, postcerebrovascular event, postheart valve repair/replacement.

Participants will be excluded if they meet one or more of the following:

1. Presence of significant risk or safeguarding concerns (ie, exercise risk).
2. Head injury/organic impairment.
3. Significant communication and/or social difficulties.

Only patients with a formal diagnosis or under assessment for one of the exclusion criteria will be excluded from the study.

## Recruitment and allocation

Participants will be screened for eligibility during routine clinic appointments by clinicians and/or cardiac team members. Clinicians/cardiac team members will ask eligible patients and their parent/caregiver if they are interested in hearing more about the research study and if their contact details can be passed onto the research team. The research team will then contact eligible and interested patients and their caregivers to provide further information about the study, answer any questions, arrange for consent to be obtained and collect baseline data. Once baseline data have been collected, participants will be randomised to a study arm using sealed-envelope software.[48] Randomisation (at the individual level) will be independent and concealed, after stratification by age (11–13 and 14–16) and sex (male and female). Within strata, allocation will occur within blocks of size 4 or 6, determined at random and with equal probability by the software. This will minimise future allocation 'predictability'. A member of the research team not blinded to allocation will contact parents/caregivers to inform them of the allocation within 2 days. Blinding of allocation will be maintained for quantitative research assistants (RAs), and the trial statistician until all outcome measures for all subjects have been collected. The trial executive committee (TEC) will regularly monitor any unblinding's and corrective action will be implemented if needed.

### Trial conditions
### CA programme

The programme will consist of six face-to-face group-based sessions lasting up to 90 min of a structured exercise programme with integrated educational and lifestyle modules, and a psychological component delivered at the recruiting hospital sites. Exercise sessions have been reviewed by a physiotherapist with

| | STUDY PERIOD | | | | | | |
|---|---|---|---|---|---|---|---|
| | Enrolment | Baseline | Allocation | Post-allocation (weeks) | | | |
| **TIMEPOINT** | Before starting CR | | | 1 | 10 | 12 | 24 |
| **ENROLMENT:** | | | | | | | |
| **Eligibility screen** | X | | | | | | |
| **Informed consent** | X | | | | | | |
| **Allocation** | | | X | | | | |
| **INTERVENTIONS:** | | | | | | | |
| **Usual Care** | | | | ←——————→ | | | |
| **CR** | | | | ←——————→ | | | |
| **ASSESSMENTS:** | | | | | | | |
| *Demographic Information Questionnaire (P)* | | X | | | | X | X |
| *SDQ (P)* | | X | | | | X | X |
| *SDQ (CYP)* | | X | | | | X | X |
| *PedsQol (CYP)* | | X | | | | X | X |
| *6MWT (CYP)* | | X | | | | X | X |
| *ISWT (CYP)* | | X | | | | X | X |
| *Physical Activity Monitoring (CYP)* | | X | | | | X | X |
| *MCQ-A (CYP)* | | X | | | | X | X |
| *CHU-9D (CYP)* | | X | | | | X | X |
| *SUQ (P)* | | X | | | | X | X |

**Figure 2** Schedule of events. Schedule of enrolment, interventions and assessments. CHU-9D, Child Health Utility-9D; CR, cardiac rehabilitation; CYP, child and young person rated; ISWT, Incremental Shuttle Walk Test; MCQ-A, Metacognition Questionnaire-Adolescent; P, parent rated; PedsQol, Paediatric Quality of Life; SDQ, Strength and Difficulties Questionnaire; SUQ, Service-Use Questionnaire; 6MWT, 6 min walk test.

experience working with patients with heart conditions and a child cardiologist who have deemed the exercise safe for CYP to engage in. Patients are free to stop the exercise if they begin to feel unwell. Lifestyle modules will focus on modifiable risk factors. The psychological treatment component (MCT) is integrated throughout each session and focuses on controlling worry and using adaptive attention strategies. Sessions include group discussions, experiential learning and homework tasks that participants are asked to complete between sessions. Participants in this treatment arm will also receive routine clinical outpatient management (TAU) alongside CR.

**Treatment as usual**
TAU will consist of routine clinical outpatient management, routine screening and medication management, which includes a review with their cardiologist or cardiac specialist nurse every 6–12 months, depending on the nature and severity of the heart condition.

**Staff training and supervision**
The intervention will be codelivered by two cardiology staff members. Staff from the cardiology department will receive two full training days in the intervention followed by 6 weeks of supervised practise in delivering the intervention to a pilot group of volunteers. A further workshop will address any challenges the staff's experienced

when delivering the intervention. Staff's delivery of the intervention will be monitored by listening to audio-recordings of sessions and they will receive supervision throughout the trial. Adherence to the intervention will be assessed using an adherence checklist completed by the staff at each session.

## Data collection

The planned study recruitment date is March 2024, with the end of follow-up marking study completion in June 2025. There will be three follow-up assessment points: baseline, 12 weeks (end of treatment) and 24 weeks (follow-up). To ensure completeness of the questionnaires, participants will be asked to arrange a visit, either at an NHS site or in their own home, at each time point. Participants' total time involvement in the study will be 24 weeks. Participants will be reimbursed with a £10 shopping voucher for completing each assessment.

## Criteria for discontinuation

Participants may withdraw from the study at any time, without any consequences to themselves, their healthcare or their ability to take part in future research. Participants who withdraw will continue to receive usual care. Participants can also be withdrawn at the request of the chief investigator or cardiology service, but this would only happen if a participant's life or long-term health or welfare was at risk from continued participation in the study.

## Outcomes

The primary outcomes of the study are feasibility and acceptability outcomes. Feasibility will be assessed using referral rates, recruitment (ie, number of patients consented and randomised, number of patients declined the study) and retention rates (ie, number of patients withdrawn from the study, lost to follow-up) and attendance at the intervention. Qualitative semistructured focus groups or interviews, as preferred, will be used to understand CYP, caregivers and healthcare professionals' views of acceptability of the intervention and targets for improvement. All outcomes listed below will be obtained at each assessment time point.

## Measures

### Strength and Difficulties Questionnaire

The Strength and Difficulties Questionnaire (SDQ)[49] is a behavioural measure completed by the caregiver and child for children aged 2–17. The measure has five subscales: emotion, hyperactivity, conduct, peer relations and prosocial behaviour. Each subscale contains five items, and each item is scored 'not true', 'somewhat true' or 'completely true'. Scores for the subscales range from 0 to 10, with total difficulties score (0–40) generated by summing together the scores from all subscales except prosocial. Higher scores on the prosocial scale reflect strengths whereas higher scores across other subscales reflect difficulties.[49] The SDQ subscales have satisfactory internal consistency (mean Cronbach's alpha=0.70) and test–retest reliability ranging from 0.75 to 0.91.[50]

### Paediatric Quality of Life

The Paediatric Quality of Life (PedsQol)[51] consists of 23 items, which measures health-related quality of life including physical, emotional, social and school functioning, and is commonly used in paediatric cardiology.[5 8] The items are scored on a Likert scale ranging from 0 (never) to 4 (almost always). Total scores range from 0 to 100, with higher scores indicating better quality of life. The PedsQol will be completed by CYP.

### Six minute walk test

The 6 minute walk test (6MWT)[52] is used to assess aerobic capacity and endurance. Participants are asked to walk as far as possible for 6 minutes and the total distance covered during this time is used as the outcome. Performance on the 6MWT has been predictive of morbidity and mortality whereby poorer performance is associated with increased mortality.[52]

### Incremental Shuttle Walk Test

The Incremental Shuttle Walk Test (ISWT)[53] is used to assess exercise capacity. The ISWT is a 12-level test (1 min in each level), whereby participants walk up and down a 10 m course with increasing acceleration of 0.17 m/s up to a maximum speed of 2.37 m/s. The walking speed is dictated by an audio signal. The test ends when the subject has a heart rate greater than 85% of their predicted maximum, is limited by dyspnoea, or when the subject is unable to maintain the required speed and does not complete a shuttle for a second consecutive time. The distance covered from the number of shuttles will be calculated.

### Physical activity monitoring

Participants will wear an accelerometer (Actigraph) for 5 days to monitor physical activity levels at each assessment. Data collected via the Actigraph include total movement, moderate to vigorous physical activity, non-sedentary time, step count and energy expenditure.

### Metacognition Questionnaire-Adolescent

The Metacognition Questionnaire-Adolescent (MCQ-A)[54] assesses metacognitive beliefs (beliefs about thinking) across five subscales: uncontrollability and dangerousness of worry, need to control, cognitive self-consciousness, positive beliefs about worry and cognitive confidence. The 30 items are scored on a Likert scale from 1 (do not agree) to 4 (agree very much) with total scores ranging from 30 to 120 and 6–30 for each subscale.[54] The internal consistency of the MCQ-A across total scores and most subscales have been supported with adequate to excellent Cronbach alphas (0.76–0.92).[55] This MCQ-A will be completed by CYP.

### Child Health Utility-9D

The Child Health Utility-9D (CHU-9D)[56] measures PedsQol using nine dimensions (including worried, sad, pain, tired, annoyed, schoolwork/homework, sleep, daily routine and ability to join in activities) and is suitable for CYP aged between 7 and 17 years. Each item has five levels ranging from no problems to inability to do the item.[56] The CHU-9D has strong internal consistency (Cronbach's alpha=0.79) and strong test–retest reliability (interclass correlation coefficients=0.75).[57] The CHU-9D will be completed by CYP.

### Demographic information questionnaire

The demographic questionnaire will collect variables including child's age, sex, weight, height, body mass index, type of heart problem, comorbid mental and physical illnesses, ethnicity, medication, socioeconomic status, parental occupational status. This will be completed by parents/primary caregivers.

### Health and Social Care Service-Use Questionnaire

The Service-Use Questionnaire will include questions about whether the child has used any primary, secondary or community-based health and social care and how often they used the service in the last 16 weeks (baseline study visit) or since the last assessment (follow-up study visits). The questionnaire will be developed from existing child-relevant service use questionnaires held by the coapplicants and through discussion with the patient and public involvement (PPI) representative, parent advisory group and clinical members of the study team. This will be completed by parents/primary caregivers.

### Qualitative Nested Study

We will investigate patients', caregivers' and healthcare professionals' experiences of the CA programme, targets for improvement, perceived acceptability of components of CR and examine participants experience of living with/managing heart conditions using semistructured interviews with CYP (n=20), caregivers (n=10) and healthcare professionals (n=10). Sampling will be purposive to ensure variation in key variables including age, type of health condition, relationship to CYP and discipline of healthcare professional (eg, physiotherapist, cardiologist, clinical psychologist). We will endeavour to recruit patients with a range of experiences of CR including those for whom engagement has been low as well as those who have attended regularly. We will explore the acceptability of CR, factors influencing participants' engagement, and barriers and enablers to CR being implemented within services. We will also explore with participants their experiences of key trial processes (eg, outcome measures, recruitment and randomisation procedures) to inform the design of a definitive trial. An interview topic guide will be used in order to maintain a consistent structure. Members from the PPI group will be involved in the development of the interview schedule and piloting for the qualitative interviews. Participants will be reimbursed for taking part in qualitative interviews or focus groups.

### Sample size calculation

Feasibility trials are not powered to provide a definitive effectiveness analysis. Therefore, the sample size is based on having sufficient patients to evaluate the acceptability/feasibility of the intervention (as measured by patient acceptability and adherence ratings, recruitment, and retention rates), and to obtain a provisional estimate of the 'promise' of the intervention (eg, an 80% CI) for powering a future definitive randomised controlled trial (RCT). We will recruit 50 patients per arm (total N=100), which would allow for an overall attrition rate of up to 20% to be determined to be within eight absolute percentage points with 95% confidence. The total sample size is in line with recommendations by Lancaster et al.[58] This will also be sufficient to estimate key 'clinical' parameters, such as the SD of potential outcomes (in the larger trial), with adequate precision, for which samples of 40 are generally sufficient.[59]

Our qualitative sample has been informed by prior experience and published guidance.[60] Previous research suggests that a sample size of 12 can achieve data saturation.[61] The exact number will be determined by data sufficiency, achieving adequate variance in recruitment and thematic saturation, however, we are confident this estimate will be sufficient to answer the relevant research questions.

### Analyses
#### Quantitative analyses

As this is a feasibility study, we will not be carrying out hypothesis testing to determine if the intervention is effective. Instead, descriptive data on study processes and clinical outcomes will be reported. Following an intention-to-treat protocol, we will present data relating to participant recruitment, randomisation, retention at follow-up and drop-out rate in a CONSORT flow chart. The attrition rate will be calculated (with a 95% CI) and reasons for drop-out will be recorded where possible. We will also assess rates of missing data (including which elements) on specific questionnaires and whether any of the measures display floor and/or ceiling effects.

We will summarise by trial arm, as appropriate (eg, mean/SD; median/IQR; proportion/95% CI; data range), data for all potential (ie, for the larger trial) clinical primary outcome measures. We will also report 80% CIs for the trial arm coefficient from appropriate regression analyses (adjusting for age, gender and the corresponding baseline outcome) by way of investigating the promise of the CA programme. This information, the SD of the SDQ and the PedsQoL, the estimated attrition rate and recruitment-based data will be used to help inform the sample size calculation for the large-scale, definitive RCT (in addition to other (published) sources). The statistical analysis plan will be produced by MH (statistician) prior to the examination of outcomes.

## Qualitative analyses

All focus groups and interviews will be transcribed verbatim. Thematic analysis (TA)[62] and the theoretical framework of acceptability[63] will be used, allowing for both inductive and deductive coding. Members from the PPIE group will provide feedback on emerging analyses.

## Economic analyses

We will evaluate the feasibility of conducting an economic evaluation of CR, by exploring the costs and outcomes associated with CR compared with standard care. We will codesign a health-resource use questionnaire with our PPIE group to capture any changes in healthcare resource use during the study, which will be completed by CYP and their guardians. We will explore resource use from the perspective of the NHS and personal social services, as well as taking a wider perspective to examine if there are any patient out-of-pocket expenses. We will also examine the resources and their associated costs required to develop and deliver the intervention.

We will measure health outcomes using two measures, as described above: the CHU-9D and PedsQol.[64] The CHU-9D has a published value set in which to generate utility scores,[56] and therefore, quality-adjusted life-years. There is also a published mapping algorithm to transform the PedsQol to the EQ-5D.[64 65] We will examine the acceptability of both measures to the CYP and assess data completeness, to inform which questionnaire is most appropriate for use in the definitive study, or if indeed, another measure of PedsQol would be more appropriate.

We will conduct a preliminary economic evaluation using the data collected, to determine whether CR plus TAU would be cost-effective from the perspective of the NHS and Personal Social Services.

Multiple imputation and censored data analysis techniques will be used to separately impute missing observations from participants who complete follow-up, and missing follow-up data for participants lost to follow-up. The primary and sensitivity economic analyses will be controlled for key baseline covariates or characteristics. Cost-effectiveness acceptability curves will assess the likely cost-effectiveness of the intervention and uncertainty in the observed data. This cost-effectiveness analysis will be limited due to the sample size and the overall health economic work will focus on informing a definitive trial.

## Trial management and oversight arrangement

The trial is managed by a TEC, which includes those individuals responsible for the day-to-day management of the study including the chief investigator, coinvestigators and identified collaborators, principal investigators (PIs) and the study statistician. Notwithstanding, the legal obligations of the sponsor and chief investigator, the TEC has operational responsibility for the conduct of the study including monitoring overall progress to ensure the protocol is adhered to and to take appropriate action to safeguard the patients and the quality of the study. There will also be PPI groups (adult and CYP), which will meet at least quarterly and provide advice and feedback on a range of trial-related activities, for example, reviewing study documents. The end of the study will be reported to the REC within the required time frame if the study is terminated prematurely. Investigators will inform patients of any premature termination of the study and ensure that the appropriate follow-up is arranged for all involved. Following the end of the study, a summary report of the study will be provided to the REC within the required timeframe.

## Data management

Participants will be allocated a study identity code number for use on all study documents and the electronic database. The study team will make a separate confidential database for the participant's name, date of birth and study identity code to permit identification of participants enrolled in the study, for example, for follow-up. All other information will be anonymised. All study documents (including participant's written consent forms) which are to be held at the participating centres will be held in strictest confidence. Access to patient information shall be restricted to authorised persons. Data will only be available through restricted, shared areas on the secure Greater Manchester Mental Health (GMMH) and University of Manchester computer systems (password and username secured). Baseline and follow-up data, which is anonymous data, will be stored in locked filing cabinets at GMMH. These data will be entered into an electronic database for analysis purposes by study team members blind to trial arm allocation. All computers are password protected and adhere to the secure storage policies of the NHS trust and University of Manchester.

## Safety reporting

Adverse events (AEs) and serious AEs (SAEs) will be monitored throughout the study. AEs are defined as any untoward medical occurrence in a patient or study-specific intervention and which does not necessarily have a causal relationship with the intervention (ie, deterioration in cardiac health not associated with the intervention). SAEs are defined as AEs that result in death, are life-threatening or require hospitalisation. Adverse reactions (ARs) are an untoward medical or unintendical medical response in a patient to a study specific intervention which has a causal relationship with the intervention (ie, physical injury due to the intervention). Serious ARs (SARs) are defined as ARs that result in death, are life-threatening or require hospitalisation. All AEs, SAEs, ARs and SARs will be reviewed by the trial's child clinical psychologist and paediatric consultant cardiologist within 24 hours of reporting. Any AEs identified as likely to be caused by the intervention during the period of recruitment or intervention on the study will be recorded at the study site using an AE/SAE record form which will be completed by the health professionals. Study RA and study PIs will be informed, and information inputted onto a secure database. The CIs will also assess if the nature of

the event is likely to have been induced because of the study intervention or processes. Any that are deemed be serious and related to the intervention will be reported to the sponsor, the study executive committee and ethics committee within 24 hours of the event.

## Ethics and dissemination

The study received full ethical approval on 14 February 2023, from North West—Greater Manchester East Research Ethics Committee, Research Ethics Committee (REC) REC ref: 22/NW/0367, IRAS ID 319134. The trial has been registered with ClinicaTials.gov (NCT05968521) and with ISRCTN (ISRCTN50031147). The study protocol used was Version 6: 21 June 2023. Any modifications to the trial will be submitted for further ethical approval and approved changes will be documented and communicated to the REC, trial registry, executive committee and all relevant parties. The study will be conducted in accordance with the ethical principles that have their origin in the Declaration of Helsinki (1996), the principles of Good Clinical Practice and the UK Policy Framework for Health and Social Care Research (2017). GMMH acts as the sponsor for this study. As the sponsor is an NHS organisation, the NHS indemnity scheme will apply. Participating sites will be liable for clinical negligence and other negligent harm to participants taking part in the study and covered by the duty of care owed to them by the sites concerned.

Written informed consent will be obtained from all participants and their parents/caregivers. Participants will be free to withdraw from the study at any time without providing a reason or their care being impacted. All the information collected during this trial will be confidential and held in accordance with NHS Data Protection guidelines and Good Clinical Practice guidelines. Confidentiality will only be breached if participants disclose information, which may indicate that there is a risk of harm to themselves or others. Every opportunity to discuss any possible breaches of confidentiality with participants will be taken prior to informing any appropriate agencies, for example, children and adolescent mental health services, general practitioner (GP) or accident and emergency (A&E) services.

All researchers and study site staff involved with the study must comply with the requirements of the Data Protection Act 1998 regarding the collection, storage, processing and disclosure of personal information, and will uphold the Act's core principles. Audiorecordings and transcriptions of interviews will be stored on NHS drives, which are password protected and designed for the storage of confidential research material. Interviews which are transcribed will be anonymised at the point of transcription. Any third party involved with transcribing of interviews will sign a confidentiality agreement and be fully instructed in how to anonymise transcripts.

The main study results will be published in the name of the study in a peer-reviewed journal, on behalf of all collaborators. The manuscript will be prepared by a writing group, appointed from among the TEC. All participating centres and clinicians will be acknowledged in this publication. All presentations and publications relating to the study must be authorised by the study executive committee and sponsor, on whose behalf publications should usually be made. Authorship of any secondary publications resulting from the intervention will reflect the intellectual and time input into these studies and will not be the same as on the primary publication. No investigator may present or attempt to publish data relating to this study without prior permission from the TEC . The findings will also be presented at national, international and regional conferences and in public involvement events where the information from this study is relevant.

## Patient and public involvement

A youth PPI group composed of young people with lived experience of a heart condition (with and without mental health difficulties) has been created along with an adult PPI group comprised of parents, caregivers and adult heart condition patients (with and without mental health difficulties). Both PPI groups have aided in the development of the CA programme, in obtaining ethical approval, codeveloped patient-facing materials, helped to develop the patient recruitment and retention plan, have helped to develop the study name and logo, and will further assist with retention, development of the dissemination plan and with dissemination of research findings.

## DISCUSSION

CHD presents as one of the most common noncommunicable childhood diseases. While medical advances have improved survival rates, young people with heart conditions continue to face significant challenges, comorbidities, reduced quality of life and increased psychological distress. NHS England has highlighted that cardiac services must be provided in a way that allows the best outcomes for all patients, not only in reducing mortality, but also in decreasing disability and improving opportunities to lead better lives.[66]

A CR programme that integrates mental and physical healthcare has the potential to meet NHS objectives[67] which aims to ensure that CYP, their family and carers have an understanding of the patient's condition, its impact on their health and future, and an understanding of the condition's psychological, social and cultural factors in addition to service specifications for cardiac disease.[68]

The current study is limited by the sample size, and it is not powered to evaluate efficacy of the intervention. Despite this, the study will attempt to provide data on important logistical and clinical outcomes, which will help inform the power calculation for a full trial.

Despite the limitations, this study would meet service standards and fits within current priorities

for closer integration of psychological and physical health services.[19 20 69] Development and evaluation of a CR programme is an important platform for the progression of young person's cardiac services and will provide valuable qualitative and quantitative data to support the design of a future large-scale definitive trial to test the effectiveness of CA for CYP with CHD.

**Author affiliations**
[1]Research and Innovation, Greater Manchester Mental Health NHS Foundation Trust, Manchester, UK
[2]NIHR School for Primary Care Research, Centre for Primary Care and Health Services Research, School of Health Sciences, University of Manchester, Manchester, UK
[3]Health Organisation, Policy and Economics, Division of Population Health, Health Services Research and Primary Care, School of Health Sciences, Faculty of Biology, Medicine and Health, University of Manchester, Manchester, UK
[4]Manchester Centre for Health Psychology, School of Health Sciences, Faculty of Biology, Medicine, and Health, University of Manchester, Manchester, UK
[5]Health Science, University of York, York, UK
[6]Department of Pedeatric Cardiology, Manchester University NHS Foundation Trust, Manchester, UK
[7]Shrewsbury and Telford Hospital NHS Trust, Shrewsbury, UK
[8]Division of Psychology and Mental Health, School of Health Sciences, Faculty of Biology, Medicine and Health, The University of Manchester, Manchester, UK
[9]Greater Manchester Mental Health NHS Foundation Trust, Manchester Academic Health Science Centre, Manchester, UK

**Contributors** LC is the chief investigator and conceived the study, designed, and developed the initial trial protocol, developed the trial intervention, and contributed to the first and subsequent drafts of the manuscript. AW is the co-chief investigator and contributed to the study conception, designed, and developed the initial trial protocol, developed the trial intervention, and contributed to the first and subsequent drafts of the manuscript. AW provided translational guidance in applying the metacognitive model in CR. MH contributed to the study design, statistical analysis and sample size and wrote the quantitative evaluation section of the manuscript. EM developed the health economics components of the protocol and wrote these sections of the manuscript. GC contributed to the development of the trial recruitment strategy. PJD contributed to study design. JM contributed to adverse event reporting. SP developed the qualitative components of the protocol and wrote these sections of the manuscript. All authors edited the manuscript and read and approved the final version.

**Funding** The study is funded by the National Institute for Health Research (NIHR) under its Research for Patient Benefit Grant (Grant Reference Number: NIHR203634) and NIHR Manchester Biomedical Research Centre (Grant Reference Number: NIHR203308). The views expressed are those of the author(s) and not necessarily those of the NIHR or Department of Health and Social Care.

**Disclaimer** The views expressed are those of the author(s) and not necessarily those of the NIHR or Department of Health.

**Competing interests** AW is the creator of Metacognitive Therapy and codirector of the MCT Institute.

**Patient and public involvement** Patients and/or the public were involved in the design, or conduct, or reporting, or dissemination plans of this research. Refer to the Methods section for further details.

**Patient consent for publication** Not applicable.

**Provenance and peer review** Not commissioned; externally peer reviewed.

**ORCID iDs**
Lora Capobianco http://orcid.org/0000-0001-6877-8650
Patrick Joseph Doherty http://orcid.org/0000-0002-1887-0237

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
