## [Reviewer comments · BMJ Open]

ARTICLE DETAILS

TITLE (PROVISIONAL)	Cardiac Rehabilitation for Children and Young People (CardioActive): Protocol for a Single-Blind Randomized Feasibility and Acceptability Study of a centre based cardiac rehabilitation programme versus usual care in 11-16-year-olds with heart conditions.
AUTHORS	Capobianco, Lora; Hann, Mark; McManus, Emma; Peters, Sarah; Doherty, Patrick Joseph; Ciotti, Giovanna; Murray, Joanne; Wells, Adrian

VERSION 1 – REVIEW

REVIEWER	Cifra, Barbara The Hospital for Sick Children
REVIEW RETURNED	26-Sep-2023

GENERAL COMMENTS	The manuscript presents the CardioActive trail protocol, a single-blind randomized feasibility and acceptability study of a cardica rehabilitation program versus usual care in 11-16-year olds with heart conditions. I read the manuscript with great interest. Below my main comments: Abstract: well-written and explanatory. Introdcution: at line 77 I belive a connecting statement is missing between the reference to adult CR and the lack of pediatric programs. Methods: 1. Eligibility Criteria: no mention in the inclision or exclusion criteria about patients being able to exercise. Please clarify. The statement on line 136 is not fully clear, please clarify. Trial condition: some important information are missing. Please clarify is the exercsie group session are onsite o remote, if sessions will be supervised or not and how compliance to the exercise protocol will be monitored. This is important for the primary outcomes. It would be important for the reader and future replicability of data if authors can clarify how the data from the 6MWT and the shuttle walk test have been used to create the exercise program and how the protocol was adapted to patients with different functional capacity. Authors mentioned alos the use of accelerometers but no data from this evaluation are mentioned as part of the outcomes. Staff training and supervision: please clarify the expertise of the training staff, specially for meta cognition. Criteria for discontinuation: Please define adverse eventss since this is an important information for any exercise intervention. Minor comment:
--

	1. Please make sure all abbreviations are properly mentioned and cited in the text. 2. Please check for proper spelling. In the title the word program is spelled "programme".
--	--

REVIEWER	Khoury, Michael University of Alberta
REVIEW RETURNED	29-Sep-2023

GENERAL COMMENTS	Thank you for the opportunity to review this study protocol of a single blind parallel randomized feasibility trial comparing a cardiac rehabilitation program (CardioActive) against usual care in 11-16 year old children with heart disease. The authors have devised a protocol (CA) that is 6-sessions and incorporates physical and mental health components. This is an interesting trial and I commend the authors for undertaking this work. However, I did have some concerns regarding the detail to which some of the methodology is Methodology:  - Why did the authors choose to include children with ALL forms of congenital heart disease? I would imagine the impact of such a program would be different in a patient with a small ASD or a bicuspid aortic valve compared to someone with Fontan circulation or post tetralogy of Fallot repair. Similarly, are all forms of cardiac arrhythmia included? There again is a wide spectrum and it is not clear if children with even very mild disease (such as occasional isolated premature ventricular contractions) would be included. - Will the baseline evaluation also be used to evaluate safety for exercise? - The details regarding the CardioActive programme is lacking somewhat. It was not clear to me if this program will take place at home or on site, if there will be any live interaction, what the monitoring will be like, etc. - Will randomization also occur by disease severity, type, etc? If not, why not? - In one part of the methodology, it indicates that the intervention is 16 weeks but elsewhere it says 12 weeks. This should be clear. - The specific criteria that will be used to define feasibility should be included and these values should be justified. - Are there not secondary outcomes to look at changes in physical and mental health parameters? I would imagine this would be reasonable to include in a study of 100 participants. - Why did the authors not choose to use CPET as an outcome measure. CPET is commonly used in exercise/CR trials. - More details regarding the accelerometer wear is needed. For example, specific measures, definitions, outcomes. - Details regarding the qualitative methodology are also lacking. I note that potential limitations or troubleshooting is not discussed in a great deal. Figure 1  - Will the baseline assessment occur prior to following randomization? This is not clear from Figure 1 Additional points:  - Some elements of the introduction could be worded better. For example, the sentence "This is despite previous systematic
--

	reviews..." (line 78) does not really make sense following the previous sentence. - The authors do not introduce MCT and spell it out. It first appears in the final paragraph of the Introduction
--	--

VERSION 1 – AUTHOR RESPONSE

Reviewer: 1

Dr. Barbara Cifra, The Hospital for Sick Children

Comments to the Author:

26 September 2023

Please find below my comments.

The manuscript presents the CardioActive trail protocol, a single-blind randomized feasibility and acceptability study of a cardiac rehabilitation program versus usual care in 11-16-year olds with heart conditions. I read the manuscript with great interest. Below my main comments:

Abstract: well-written and explanatory.

Response: Thank you.

Introduction: at line 77 I believe a connecting statement is missing between the reference to adult CR and the lack of pediatric programs.

Response: We have revised the paragraphs at line 84 and 85 to provide a better link between adult CR and lack of pediatric programs.

Methods:

1. Eligibility Criteria: no mention in the inclusion or exclusion criteria about patients being able to exercise. Please clarify. The statement on line 136 is not fully clear, please clarify.

Response: In the eligibility criteria it states significant risk or safeguarding concerns. Significant risk includes those individuals who would be too high risk to take part in exercise. We have listed this on page 6.

Trial condition: some important information are missing. Please clarify if the exercise group session are onsite or remote, if sessions will be supervised or not and how compliance to the exercise protocol will be monitored. This is important for the primary outcomes. It would be important for the reader and future replicability of data if authors can clarify how the data from the 6MWT and the shuttle walk test have been used to create the exercise program and how the protocol was adapted to patients with different functional capacity.

Response: On page 7 we have added that sessions are delivered face to face at the hospital site. On page 8 under the heading staff training and supervision we state that adherence to the intervention will be assessed using an adherence checklist to be completed by staff and that sessions will be audio recorded. We have added that the intervention will be co-delivered by two cardiology staff members, note that more than 2 staff members will be trained to deliver the intervention but only 2 staff members at a time are required so children are supervised throughout. We appreciate the reviewers comment regarding adaptations of the intervention, we intend to publish an article describing the development of the intervention of which that paper will include the details requested on adaptations for varying functional capacity of children.

Authors mentioned also the use of accelerometers but no data from this evaluation are mentioned as part of the outcomes.

Response: Accelerometers collect a vast amount of data as such we have opted not to list all of the data that can be collected with such devices. We have however included some examples of the data that can be collected. We now state on page 10, Data collected via the Actigraph includes total movement, moderate to vigorous physical activity, non-sedentary time, step count, and energy expenditure.

Staff training and supervision: please clarify the expertise of the training staff, specially for meta cognition.

Response: Staff are not required to have any prior training or experience in mental health. This will be covered as part of the training. Our research group has previously and successfully trained cardiac rehabilitation staff (i.e. nurse, physiotherapists and occupational therapists) in adult services to deliver metacognitive therapy in adult cardiac rehabilitation services. See the following papers for further details:

Criteria for discontinuation: Please define adverse eventss since this is an important information for any exercise intervention.

Response: We have expanded our definition of adverse and serious adverse events on page 16 and provided some examples which now states AEs are defined as any untoward medical occurrence in a patient or study-specific intervention and which does not necessarily have a causal relationship with the intervention (i.e., deterioration in cardiac health not associated with the intervention). SAEs are defined as AEs that result in death, are life-threatening or require hospitalisation. Adverse reactions (ARs) are an untoward medical or unintended medical response in a patient to a study specific intervention which has a causal relationship with the intervention (i.e., physical injury due to the intervention). Serious adverse reactions (SARs) are defined as ARs that result in death, are life-threatening or require hospitalisation. All AEs, SAEs, ARs and SARs will be reviewed by the trial's child clinical psychologist and paediatric consultant cardiologist within 24 hours of reporting.

Minor comment:

1. Please make sure all abbreviatoons are properly mentioned and cited in the text.

Response: Thank you we have reviewed the paper to ensure all abbreviations are fully explained.

2. Please check for proper spelling. In the title the word program is spelled "programme".

Response: Thank you the spelling of programme is consistent with the UK spelling and location of where the project is based.

Reviewer: 2

Dr. Michael Khoury, University of Alberta

Comments to the Author:

Thank you for the opportunity to review this study protocol of a single blind parallel randomized feasibility trial comparing a cardiac rehabilitation program (CardioActive) against usual care in 11-16 year old children with heart disease. The authors have devised a protocol (CA) that is 6-sessions and incorporates physical and mental health components. This is an interesting trial and I commend the authors for undertaking this work. However, I did have some concerns regarding the detail to which some of the methodology is

Methodology:

- Why did the authors choose to include children with ALL forms of congenital heart disease? I would imagine the impact of such a program would be different in a patient with a small ASD or a bicuspid aortic valve compared to someone with Fontan circulation or post tetralogy of Fallot repair. Similarly, are all forms of cardiac arrhythmia included? There again is a wide spectrum and it is not clear if children with even very mild disease (such as occasional isolated premature ventricular contractions) would be included.

Response: We have opted to take on a range of heart conditions as all heart conditions require exercise. The inclusion criteria also mirrors the broad inclusion criteria used within adult cardiac rehabilitation services and allows a greater number of patients able to take part in such a programme. While all forms of arrhythmia may be included it is up to the referring consultant cardiologist to also deem the patient eligible for this programme, consultants may decide that they feel their patient would be ineligible for this type of programme and may be an important factor in understanding clinicians decision making and acceptability/feasibility of the types of patients referred to the programme.

- Will the baseline evaluation also be used to evaluate safety for exercise?

Response: The exercise programme is graded, as such the baseline evaluation will also serve to assess which format of the exercises children should start at.

- The details regarding the CardioActive programme is lacking somewhat. It was not clear to me if this program will take place at home or on site, if there will be any live interaction, what the monitoring will be like, etc.

Response: We have added further details on the location of the programme and that it will take place face to face. We have also added details under the staff training and supervision section that the intervention will be co-delivered by two cardiology staff who will supervise the programme. We intend to publish a subsequent paper on the details on the intervention development.

- Will randomization also occur by disease severity, type, etc? If not, why not?

Response: Randomization will using stratification by age (11-13; 14-16) and sex (male; female). Due to the range of CHD types we have opted not to include disease type of severity as a randomization factor. As the study is feasibility and acceptability this may be included in the future.

- In one part of the methodology, it indicates that the intervention is 16 weeks but elsewhere it says 12 weeks. This should be clear.

Response: The intervention is 6 weeks but the follow up time frame is 12 weeks. We have reviewed the manuscript to ensure that this is clear throughout.

- The specific criteria that will be used to define feasibility should be included and these values should be justified.

Response: Values for markers of feasibility are generally considered standard when following the traffic light criteria such that >80% (green = proceed to full trial); 60-80% (amber = refine prior to full trial); and <60% (red = do not proceed to full trial). These will apply to participants recruitment and retention.

While other feasibility outcomes will be assessed such as attendance at CR and referral rates the purpose of the feasibility trial is to assess these variables in order to determine what amendments may need to be made ahead of a trial.

- Are there not secondary outcomes to look at changes in physical and mental health parameters? I would imagine this would be reasonable to include in a study of 100 participants.

Response: As stated on page 9 we will include the strength and difficulties questionnaire and pediatric quality of life measure, the incremental shuttle walk test, six minute walk test, activity monitoring. Outcomes include height, weight, BMI, and resting heart rate will also be included.

- Why did the authors not choose to use CPET as an outcome measure. CPET is commonly used in exercise/CR trials.

Response: We opted not to use CPET as it is known to increase participant burden, is disliked by some children and in 40% of tests in children with congenital heart disease it fails to yield a valid exercise capacity (RER >1.0)*. Exercise capacity in this population is also negatively correlated with disease severity as measured by the New York Heart Association classification.**

*van Genuchten, W.J., Helbing, W.A., Ten Harkel, A.D.J. et al. Exercise capacity in a cohort of children with congenital heart disease. *Eur J Pediatr* 182, 295–306 (2023).

<https://doi.org/10.1007/s00431-022-04648-9>

**Mackenzie Buchanan, Christopher Spence, Michelle Keir, Michael Khoury. Physical Activity Promotion Among Individuals With Tetralogy of Fallot. *CJC Pediatric and Congenital Heart Disease*, 2023, ISSN 2772-8129, <https://doi.org/10.1016/j.cjpc.2023.08.002>.

The aim of our intervention is to promote high repetition aerobic and part-body weight resisted physical movements in a fun setting to help build confidence and stamina in children who are known to avoid physical activity.**

The intervention duration is six weeks and done within the child's own perception of moderate to vigorous exertion cross checked by staff supervising the exercise sessions. which is unlikely to increase exercise capacity as defined by CPEX.

- More details regarding the accelerometer wear is needed. For example, specific measures, definitions, outcomes.

Response: The accelerometer is worn for 5 days at each assessment time point. The accelerometer is a wGT3X-BT by Actigraph. The accelerometer will collect data on outcomes including: total movement, moderate to vigorous physical activity, non-sedentary time, step count, and energy expenditure.

- Details regarding the qualitative methodology are also lacking.

Response: It is unclear what additional details the reviewer would like us to add. We have described the participants being recruited, methodology, aims of the qualitative interviews, and how the interviews will be analysed. We feel we currently provide sufficient details on the qualitative methods.

I note that potential limitations or troubleshooting is not discussed in a great deal.

Response: We have added in limitations to the discussion.

Figure 1

- Will the baseline assessment occur prior to following randomization? This is not clear from Figure 1

Response: Yes, baseline assessment occurs prior to randomization.

Additional points:

- Some elements of the introduction could be worded better. For example, the sentence "This is despite previous systematic reviews..." (line 78) does not really make sense following the previous sentence.

Response: We have revised line 78 to provide a better link to the subsequent paragraph and in doing so revised this sentence.

- The authors do not introduce MCT and spell it out. It first appears in the final paragraph of the Introduction

Response: We have revised the introduction to focus and provided a more detailed overview of the metacognitive model to provide further background on the theory behind in the psychological approach.

VERSION 2 – REVIEW

REVIEWER	Khoury, Michael University of Alberta
REVIEW RETURNED	10-Dec-2023
GENERAL COMMENTS	The authors have adequately addressed all my prior comments